# Sensitivity and Specificity for the Detection of Clinical Mastitis by Automatic Milking Systems in Bavarian Dairy Herds

**DOI:** 10.3390/ani12162131

**Published:** 2022-08-19

**Authors:** Mathias Bausewein, Rolf Mansfeld, Marcus G. Doherr, Jan Harms, Ulrike S. Sorge

**Affiliations:** 1Bavarian Animal Health Services, 85586 Poing-Grub, Germany; 2Clinic for Ruminants with Ambulatory and Herd Health Services, Centre for Clinical Veterinary Medicine, LMU Munich, 85764 Oberschleissheim, Germany; 3Institute for Veterinary Epidemiology and Biostatistics, Freie Universität, 14163 Berlin, Germany; 4Institute for Agricultural Engineering and Animal Husbandry, Bavarian State Research Centre for Agriculture, 85586 Poing-Grub, Germany

**Keywords:** dairy cow, automatic milking system, clinical mastitis detection

## Abstract

**Simple Summary:**

This cross-sectional study assessed the sensitivity and specificity of clinical mastitis detection by automated milking systems on Bavarian dairy herds in southern Germany. Clinical mastitis affects animal health and food safety, and therefore, its detection is an important task of any automatic milking system. Different manufacturers have different approaches to detecting clinical mastitis, with detection rates (sensitivity) ranging between 31% and 78% and correct rejection rate (specificity) between 79% and 97%. In multivariable models, some cow-level factors were shown to influence these rates.

**Abstract:**

In automatic milking systems (AMSs), the detection of clinical mastitis (CM) and the subsequent separation of abnormal milk should be reliably performed by commercial AMSs. Therefore, the objectives of this cross-sectional study were (1) to determine the sensitivity (SN) and specificity (SP) of CM detection of AMS by the four most common manufacturers in Bavarian dairy farms, and (2) to identify routinely collected cow data (AMS and monthly test day data of the regional Dairy Herd Improvement Association (DHIA)) that could improve the SN and SP of clinical mastitis detection. Bavarian dairy farms with AMS from the manufacturers DeLaval, GEA Farm Technologies, Lely, and Lemmer-Fullwood were recruited with the aim of sampling at least 40 cows with clinical mastitis per AMS manufacturer in addition to clinically healthy ones. During a single farm visit, cow-level milking information was first electronically extracted from each AMS and then all lactating cows examined for their udder health status in the barn. Clinical mastitis was defined as at least the presence of visibly abnormal milk. In addition, available DHIA test results from the previous six months were collected. None of the manufacturers provided a definition for clinical mastitis (i.e., visually abnormal milk), therefore, the SN and SP of AMS warning lists for udder health were assessed for each manufacturer individually, based on the clinical evaluation results. Generalized linear mixed models (GLMMs) with herd as random effect were used to determine the potential influence of routinely recorded parameters on SN and SP. A total of 7411 cows on 114 farms were assessed; of these, 7096 cows could be matched to AMS data and were included in the analysis. The prevalence of clinical mastitis was 3.4% (239 cows). When considering the 95% confidence interval (95% CI), all but one manufacturer achieved the minimum SN limit of >80%: DeLaval (SN: 61.4% (95% CI: 49.0%–72.8%)), GEA (75.9% (62.4%–86.5%)), Lely (78.2% (67.4%–86.8%)), and Lemmer-Fullwood (67.6% (50.2%–82.0%)). However, none of the evaluated AMSs achieved the minimum SP limit of 99%: DeLaval (SP: 89.3% (95% CI: 87.7%–90.7%)), GEA (79.2% (77.1%–81.2%)), Lely (86.2% (84.6%–87.7%)), and Lemmer-Fullwood (92.2% (90.8%–93.5%)). All AMS manufacturers’ robots showed an association of SP with cow classification based on somatic cell count (SCC) measurement from the last two DHIA test results: cows that were above the threshold of 100,000 cells/mL for subclinical mastitis on both test days had lower chances of being classified as healthy by the AMS compared to cows that were below the threshold. In conclusion, the detection of clinical mastitis cases was satisfactory across AMS manufacturers. However, the low SP will lead to unnecessarily discarded milk and increased workload to assess potentially false-positive mastitis cases. Based on the results of our study, farmers must evaluate all available data (test day data, AMS data, and daily assessment of their cows in the barn) to make decisions about individual cows and to ultimately ensure animal welfare, food quality, and the economic viability of their farm.

## 1. Introduction

In the last decades, advances in technology and automation have impacted many aspects of modern dairy farming [1]. The promise of reduced workload and more flexible work hours as a quality-of-life benefit is encouraging more and more farmers to switch to an automatic milking system (AMS), despite an initially higher economic burden [2,3]. In Bavaria, a southern region of Germany, the number of AMS farms has more than quadrupled in the last ten years [4]. 

Regardless of the milking system, mastitis remains a frequent and costly disease on dairy farms [5,6,7]. In its clinical manifestation, this inflammation of the udder often results in visible milk changes such as flakes, clots, pus, or watery milk [8]. As abnormal milk is unfit for human consumption, milk producers are required, according to EU Regulation 853/2004, to assess the milk organoleptically or with an equivalent method at each milking [9]. Besides the implications for food safety, the accurate detection of clinical mastitis (CM) allows targeted treatments of sick animals and is therefore essential to ensure animal welfare [10]. Unlike conventional milking systems, the inspection, assessment and, if necessary, the decision to separate milk has to be done automatically by the AMS [11]. For this reason, AMSs are equipped with various sensor systems to detect abnormal milk. These raw sensor data are analyzed and summarized as warning lists for the farmer in the respective herd management software [12]. However, no direct alert is given for clinical mastitis; the list will include a more general indication of a potential udder health problem. Different approaches are currently used to detect and process changes in the milk. The commercially available sensor systems provide, for example, information on electrical conductivity (EC), somatic cell count (SCC), milk yield (MY), and milk color as well as inflammation-indicating enzymes.

The most common sensor system in AMSs is the measurement of milk EC [13,14]. During CM, changes in milk ion concentrations can be observed due to increased vascular permeability caused by the inflammatory response [15,16]. However, fluctuations of ion concentrations also occur in the presence of non-disease-related influences, such as lactational period [17,18]. This may lead to inadequate detection of CM based on EC alone [19,20]. Another sensor technique gaining importance in AMSs is the in-line measurement of SCC. These sensor systems use a defined milk sample volume and can determine SCC by either automated counting of stained, fluorescent nuclei [21] or by automated CMT [22]. Despite some non-infectious-related influences on SCC, such as breed [23,24], it is a commonly used gold standard for detecting udder health problems [13]. In addition to a drop in MY due to CM [25,26], the assessment of milk color, measured by light reflectance or transmission, is another means to detect abnormal milk [27,28]. However, this information alone is not suitable for CM detection due its dependence on milk fat content [29]. Therefore, in order to improve the monitoring of udder health, new sensors have been introduced in recent years, such as the measurement of inflammatory enzymes such as L-lactate dehydrogenase [30,31], or a physical sensor such as infrared thermography that determines the temperature of an inflamed udder [32]. The combination of sensor data can lead to improvements in CM detection [33,34,35]. While the performance of mastitis detection in AMSs has been widely studied in recent years, the detection of clinical mastitis of different AMS types has hardly been considered in the field. Different approaches, study populations, and gold standard definitions further resulted in a variety of performance indices for mastitis detection in AMSs [28,36]. In the context of animal welfare and food quality assurance, however, a satisfactory detection of CM irrespective of the system is essential. 

Therefore, the first objective of this study was to determine the sensitivity (SN) and specificity (SP) of CM detection by AMSs from the four most common manufacturers in Bavarian dairy farms. The second objective was to identify parameters among those (i) routinely collected by the AMS at cow level and (ii) originating from the monthly testing of the regional Dairy Herd Improvement Association (DHIA) that, when incorporated into a multivariable model, could improve the SN and SP of clinical mastitis detection.

## 2. Materials and Methods

### 2.1. Study Design and Herd Selection 

For our study, the free web-based sample size calculator EpiTools formulas for estimating a single proportion with a given precision were used to calculate the sample size [37]. To estimate an assumed specificity of 99% with a precision of +/−1% and a confidence of 95%, at least 321 healthy cows from a finite population (N = 10,000 cows) had to be included in the sample. Our cross-sectional design did not allow selection of cows with regards to clinical udder health status. Therefore, the actual sample size was based on the expected number of clinical mastitis cases, which served as the gold standard for sensitivity estimation. We assumed that, on average, 2% of cows in a herd would have CM at any given point in time. To assess whether an 80% sensitivity was reached with assumed true prevalence of 2% and desired precision of +/−1%, approximately 2000 cows would need to be screened per AMS manufacturer to obtain 40 cows with CM. Given that one milking robot milks on average about 60 cows [38], at least 30 AMS per manufacturer were needed. The four most common AMS manufacturers in Bavaria, (in alphabetical order) DeLaval (DeLaval International AB, Tumba, Sweden), GEA Farm Technologies (GEA Farm Technologies, Bönen, Germany), Lely (Lely Industries N.V., Maassluis, The Netherlands), and Lemmer-Fullwood (Lemmer-Fullwood GmbH, Lohmar, Germany) were contacted. They provided a list of potential herds from which 30 AMSs per manufacturer were recruited. The inclusion criteria were that the selected herds maintained their AMSs regularly and preferably participated in the monthly testing by the Bavarian regional DHIA (Landeskuratorium der Erzeugerringe für tierische Veredelung in Bayern e.V., Munchen, Germany).

### 2.2. Data Collection 

Each farm was visited once between September 2019 and August 2020 by trained and specifically instructed udder health technicians of the Bavarian Animal Health Services (TGD). On each farm, first the AMS data from the herd management system was downloaded in accordance with manufacturer guidelines. Then the udder health of all lactating cows milked by the AMS was assessed in the barn. For this purpose, cows were fixed in head locks and their udder was examined for pathological changes such as redness, swelling, or hardness visually and by palpation. Foremilk from each quarter was collected on a CMT scoring plate. Pathological changes as well as the quality of abnormal milk (watery, small or large flakes etc.) were recorded and a CMT was performed. In addition, aseptic quarter milk samples were collected, and the teat end condition (score 1–4; highest score recorded per cow [39]) as well as the cow’s hygiene (score 1–4 [40]) were assessed at cow level. After the sampling, the herd management strategies, farm structure, and AMS-specific data such as cleaning, disinfection, etc. were recorded (checklist available upon request). The DHIA provided available data for the last six monthly performed test days. These included date of milk recording, test-day milk yields (kg), fat (%), protein (%), and urea concentration (ppm), as well as SCC measurements (cells/mL). The data and sample collection did not require ethical approval under German animal protection law.

### 2.3. Gold Standard Definition

Based on the findings from clinical examination of the udder, CM was recorded as grade 1 (abnormal milk with change in character, like watery or bloody, and/or the occurrence of flakes, clots, or pus of at least one quarter), grade 2 (abnormal milk in addition to local signs of inflammation of the quarter), or grade 3 (signs of grades 1 and 2 in addition to systemic signs, e.g., off feed, fever) in accord with Bradley and Green [41]. The gold-standard definition of CM in our study included grades 1–3 on a cow level, as the milk of an affected cow would have to be discarded. Cows were classified as having CM (1) or not (0). 

### 2.4. AMS Data

Commonly, the AMS warning lists about udder health include only cows that the system flagged for inconsistencies in their parameters. However, for this study we needed the information on all milking cows of the respective herd. Therefore, a full backup file of the AMS data was extracted on all farms and original lists were generated with the help of the manufacturers in two ways: The software support teams of DeLaval and GEA Farm Technologies helped us to extract the needed lists directly out of backup files with the respective AMS herd management software. Lely and Lemmer-Fullwood created lists for this study prior to the herd visit that expanded the commonly used warning lists to include healthy cows as well. These lists as well as a full backup were electronically saved at the farm visit. The list names and brief descriptions are shown in Table 1.

### 2.5. Clinical Mastitis Alert 

None of the AMS manufacturers provided a definition of CM. Instead they gave reference values for potential udder health problems. Therefore, after consultation with the manufacturers, the following markers were used as alerts for CM for this study:*DeLaval*

The mastitis detection index (MDi; DeLaval International AB, Tumba, Sweden) was used as a CM alert for DeLaval. The MDi is a mathematically generated index that considers EC and blood presence, which are both measured at quarter level, and milking interval. It uses values between 1 and 6 [42]. According to DeLaval, cows with an MDi ≥ 1.4 should be checked for udder health problems. An MDi of ≥2.0 is considered an acute warning for an udder health problem.
*GEA Farm Technologies*

The cow’s listing in the “AMS_udder_health_monitoring” list was used as the CM alert. This list is divided into three subgroups: “List1”, which displays cows that have a deviation in EC value between the quarter with the highest average EC value and the quarter with the lowest average EC value in the last four milkings. The lowest EC quarter value must be greater than 400 (manufacturer’s own unitless measurement). The factory setting for the deviation from which cows are displayed in List1 is ≥30%. “List2”, which displays cows with an EC deviation within a quarter of the default setting ≥ 110%. This list is identical to the “AMS_Increased_conductivity” list, which is checked daily. The third subgroup “Acute_ Udder_Health_warnings” summarized cows which have been flagged on both “List1” and “List2”.
*Lely*

To check udder health, the original lists Report12 and Report23 were modified to show all milking cows and highlight those cows that were normally shown on these alert lists. Report12 (“Action_list”) displays cows with a new indication for mastitis within the last 24 h, while cows remain on Report23 (“Monitor_list”) until a milking without mastitis marker occurred. Factory settings for these reports were a deviation of daily milk yield (MY) of 4.0 kg or 20% and/or a decrease of daily milk production of more than 7 kg. A 20% deviation of the EC from either the last milking and/or from the 3-day average value, as well as an absolute EC threshold of 100 (manufacturer’s own value without unit) were used as indicators of udder health problems. Milk temperature changes above a deviation factor of 2.0 as well as above the SCC threshold of 500, measured by Lely’s MQC-C system, were also considered an indication for udder health problems. Marker thresholds could be customized by farmers, but they were asked to leave them in the default settings for this study.
*Lemmer-Fullwood*

The 4QCM-System gives an indication of cows suspected of mastitis based on a quarter-level EC deviation of at least 35% from the 10-day average and/or a threshold value of >7.5 mmho (manufacturer’s own unit) for the current milking. This indication was set as the CM alert. The farmers were able to adjust these thresholds, but they were asked to leave them in the default settings for this study.

### 2.6. DHIA Data

Missing monthly test day data, e.g., during the dry period, were excluded. Then, eight new variables were generated to represent the changes in the cow’s SCC (Table 2). The DHIA data were then aggregated at the most current test day.

### 2.7. Statistical Analysis

Data arrangement and analysis was completed in SAS 9.4 (SAS Institute Inc., Cary, NC, USA). The respective manufacturers’ lists were merged at the last recorded milking. The AMS data lists were merged with the clinical observations for each cow and corresponding DHIA data. This resulted in one data set for each AMS manufacturer. Cows that were not clearly identifiable, <3 DIM, not milked for >24 h prior to backup, not milked by the AMS on the quarter that showed clinical signs of mastitis on examination, and cows that had missing CM alert values were excluded from the data set (Table 3). Data sets for each AMS manufacturer containing all available variables are available upon request. Using the abovementioned alerts for CM, sensitivity (SN) and specificity (SP) were assessed for each manufacturer individually. Binomial proportions were derived using the statement PROC FREQ with the method BINOMIAL and option EXACT. Target proportions were set for SN (*p* = 0.8) and SP (*p* = 0.99), and the alpha level of statistical significance was set to α = 0.05. 

For the second objective, two generalized logistic mixed models were used for each manufacturer to identify factors associated with SN and SP, respectively. To represent the populations for each model, observations were divided into two groups according to the occurrence of CM (as diagnosed by the technicians during the on-site visit). Thus, the SN models were run on a dataset that included only CM-positive cows, and the SP models used only data from healthy CM-negative, cows. The outcome, i.e., the binary dependent variable (coded 0/1) for the SE model was defined as true positive CM detection (1), and for the SP model as true negative CM detection (1). In a first step, quarterly individual measurements were scaled to cow level and analyzed for associations with CM status using PROC NPAR1WAY. To identify potential predictor variables for SN and SP, respectively, continuous variables and categorical variables were screened by PROC NPAR1WAY and by PROC FREQ, respectively. They were potential predictors of the multivariable model if *p* < 0.25. To avoid collinearity of the possible predictors, variables were screened using PROC CORR SPEARMAN and PROC FREQ AGREE for continuous and categorical predictors, respectively. Variables with a Spearman correlation coefficient or kappa > 0.6 were excluded. Prior to the multivariable model approach, all potential predictors were tested individually for their association with the dependent variable in a mixed logistic regression (PROC GLIMMIX) that included herd as a random effect. In order to achieve a better fit, some variables were subjected to a transformation or categorization process, e.g., the grouping of the days in milk (DIM) into <60 d, 60–120 d, >120 d. The final generalized logistic mixed model was performed with PROC GLIMMIX with option IC = Q for computation of model fit information criteria and herd as random effect. Using a manual stepwise elimination procedure, the variable with the highest *p*-value was excluded from the model after each run until all remaining variables had *p* ≤ 0.05. Then, the excluded variables were individually reentered into the model in the same order in which they were excluded to test for confounding. If a change in regression parameter estimate of ≥20% occurred in other variables, then that variable remained in the model as a potential confounder. Interactions between predictors could not be considered due to the large number of independent variables with missing biological connections. For all models, goodness of fit was assessed using the −2 Res Log Pseudo-Likelihood.

## 3. Results

Between September 2019 and August 2020, 114 dairy farms with a total of 126 AMSs were visited once by a team of two technicians from the Bavarian Animal Health Services. In total, 23 trained technicians were involved in the data-collection process. The characteristics of participating herds has been summarized in Table 4.

### 3.1. Sensitivity and Specificity of CM Alerts



*DeLaval*



For 7 of the 27 DeLaval herds, only data from the daily routine backup at 1 or 2 am could be obtained. Exclusion of these seven herds did not change the results, and therefore their data remained in the statistical analysis. Data for 1831 cows, including 70 CM cases, were available for the analysis (Table 3). The results of SN and SP for different MDi thresholds are shown in Table 5. The highest value for SN and SP was reached with an MDi threshold of 1.4 (SN: 61.4%, 95% CI: 49–72.8%; SP: 89.3%, 95% CI: 87.7–90.7).
*GEA Farm Technologies*

All backup data for the herds milking with a GEA AMS (*n* = 29) could be used for final analysis. This included data for 1616 cows, of which 54 cows were diagnosed with CM (Table 3). SN and SP of the CM alert of the AMS from the list “AMS_udder_health_monitoring” is shown in Table 5. The subgroup “List1” resulted in the highest SN of 75.9% (95% CI: 62.4–86.5) and SP of 79.2% (95% CI: 77.1–81.2).
*Lely*

The data from the AMS herd programs of all Lely study herds (*n* = 31), including 2076 cows (78 CM cases), were used in the analysis (Table 3). The SN and SP of the udder health monitoring lists provided by Lely are shown in Table 5. Use of the Lely "Monitor_list” resulted in the highest SN of 78.2% (95% CI: 67.4–86.7) with an SP of 86.2% (95% CI: 93.8–95.8).
*Lemmer-Fullwood*

Of the 27 herds provided by Lemmer-Fullwood, one backup file could not be used due to unrecoverable data. The associated herd was removed from the data set, resulting in available data for 1545 cows from 26 herds, with 37 CM cases, for statistical analysis (Table 3). In addition, some variables, such as milk lactose, milk fat, and milk protein measured by the AMS from the specifically created list “control_report_milking_10 days” could not be assigned without doubt to their respective given meaning, so that these variables were excluded from analysis. The SN and SP of the list provided by Lemmer-Fullwood for udder health monitoring “4QCM_ 10_days” are shown in Table 5. The 4QCM system achieved an SN of 67.6% (95% CI: 50.2–82) and an SP of 92.2% (95% CI: 90.8–93.5) for the detection of CM.

### 3.2. Sensitivity and Specificity Predictors



*SN Predictors*



Table 6 shows these models and their predictors by manufacturer. For DeLaval and GEA Farm Technologies, only EC could be identified as a factor that improved SN for CM detection. For Lely, with every 1 log increase in log-transformed SCC (logSCC) from Lely’s MQC-C system the odds of correctly identifying a sick animal increased (OR: 4.1; *p* = 0.002). No additional SN predictor was identified for Lemmer-Fullwood.
*SP Predictors*

Table 7 shows the four models and their predictors associated with the correct negative detection of the specified alerts from the four manufacturers. Among other predictors, the odds of being correctly classified as a healthy cow decreased with increasing milking interval (MIH, in hours) for DeLaval (OR: 0.8; *p* < 0.01) and GEA Farm Technologies (OR: 0.9; *p* < 0.01). For both Lely and Lemmer-Fullwood the EC at the quarter level (dichotomized) were also identified as helpful predictors for healthy cows. Lely cows with a quarter-level EC of less than 72 (manufacturer internal unitless score; OR: 3.73; *p* < 0.01) and Lemmer-Fullwood cows with a quarter-level-based EC below 5.6 mmho (manufacturer internal unit; OR:13; *p* < 0.01) were more likely to be correctly classified as healthy cows. For all AMS manufacturers, the udder health status based on DHIA tests was useful as a predictor for SP. Cows classified as “healthy” here had up to five times the odds of being correctly considered not to be affected by CM than cows classified as “chronic”: DeLaval (OR: 5; *p* < 0.01), GEA Farm Technologies (OR: 5; *p* < 0.01), Lely (OR: 2.2; *p* < 0.01), and Lemmer-Fullwood (OR: 5; *p* < 0.01).

## 4. Discussion

This study investigated the performance of CM detection by currently used AMSs from the four most common AMS manufacturers in Bavaria, southern Germany. The strengths of the study were a sufficiently large overall sample size, inclusion of multiple commercial farms for each manufacturer, and the use of the true gold standard for clinical mastitis, i.e., clinical observation. In addition, we were able to identify automatically recorded parameters that could improve the sensitivity and specificity of the AMS mastitis classification (“alerts”) when considered by the farmer in post-AMS analysis of the data collected at each milking.

The detection of clinical mastitis is critical for the farmer’s decision-making. For one, sick animals need to be identified to be treated or their clinical development closely observed. Furthermore, abnormal milk must not enter the food supply chain and needs to be discarded. The gold standard is the organoleptic detection of abnormal milk by the human milker. Thus, AMS must be able to guarantee the legal regulations for ensuring safe food to at least the same extent as human milkers. It has been estimated that milking technicians will find approximately 80% of CM cases going through the parlor [43]. It was therefore positive to note that the AMS of all manufacturers achieved the minimum SN of >70% as required by Annex C of ISO 20966:2007 [44]. However, the slightly higher minimum value of 80% for SN called for by Hogeveen et al. [45] was achieved by only three of four manufacturers; the SN of DeLaval fell slightly short. Since the point estimates for each manufacturer provide only an average, the confidence intervals are a better estimate of the SN range and precision. These included the required 70% SN of all AMSs and the required SN of 80% by three AMS manufacturers. However, while one might argue that the range of the intervals are fairly wide, we had enough statistical power to find potential differences, i.e., to test our hypothesis. Therefore, the identification of clinical mastitis cases has to be considered sufficient for food quality and animal welfare, especially in view of the low prevalence of the disease, i.e., 70–80% of the few cows with clinical mastitis per herd were identified. A large cost factor of CM is discarded milk [46].

Focusing on the SP, none of the evaluated systems reached the >99% SP required by ISO 20966:2007 and Hogeveen et al. [44,45]. Since the vast majority of cows in a herd will not have clinical mastitis, a farmer would suffer substantial economic losses due to falsely discarded “abnormal” milk (false positive cases) [47,48] if they use the system alerts to automatically separate milk. Whether the alert lists have a high SN or SP for subclinical mastitis was not answered in this study, since the focus of this study was clinical cases. However—purely from a legal perspective—visually normal milk in cases of subclinical mastitis does not warrant the automatic discarding of milk unless the bulk tank SCC would exceed legal limits.

Our SN and SP estimates are in agreement with previous studies [10,49]. Slight differences between studies are likely due to different gold standards, evaluated alerts, or sensors, as well as different sampling timeframes [19,28,34,50]. Brandt [51] found an SN mostly below 60% and SP above 90% for three different AMS manufacturers in 12 northern German herds, with comparable gold standard (alteration in homogeneity of the foremilk) and time window (milking right after sampling). Castro et al. [49] evaluated three different AMS types in ten Galician herds and estimated an average SN of 58% and SP of 94% for the detection of a mastitis case, based on positive CMT. Only Dalen et al. [10] found higher SN values of 80% and SP values of 90% by evaluating the online cell count (OCC) device of the DeLaval AMS operating on a Norwegian research farm, where they used veterinary mastitis treatments as gold standard.

While the results for all evaluated manufacturers were comparable, slight differences were observed and likely due to the different underlying sensor technology and proprietary algorithms. The reviewed alerts of GEA Farm Technologies and Lemmer-Fullwood are purely based on measuring EC [19]. Using EC alone is not considered sufficient for CM detection [16,20,34], due to the impact of milk temperature, fat content, or milk fractions [52,53]. Nevertheless, these alerts reached the required minimum SN thresholds—probably because not the absolute EC but the variation at quarter level was considered by system alerts. This has been shown to improve the usefulness of EC for mastitis detection [18,54]. DeLaval and Lely processed multiple sensor data into an indication of udder health problems. Although combining EC with other sensor data should improve the detection of CM [35,55], the AMS by DeLaval did not achieve the minimum SN required by Hogeveen et al. [45]. This might be because MDi is advised more as a probability of an udder health problem [35] and less for detection of abnormal milk as it occurs mainly in CM. With a higher MDi threshold, no better SN was detected in our study. This contradicts in part the approach of Lusis et al. [56], which suggested an MDi threshold of ≥2 for abnormal milk detection to keep the SCC of bulk tank milk at a low level. While the higher threshold markedly increased the SP, the SN decreased drastically. Subsequently, milk from many undetected cows with CM would still be collected. The use of detection of blood in milk as an additional sensor in MDi may be insufficient. However, in a field study by Hovinen et al. [57] every case of bloody milk could be detected, but many were detected due to the yellow color of the milk, which in turn depends on the milk fat color and thus on feed and breed [29]. Lely’s SN values are likely to be obtained by combining several sensor data and especially the overall use of SCC data in these study herds [13,21,58,59]. Data from other manufacturers’ SCC sensors could not be utilized in our study due to low numbers (e.g., only seven DeLaval herds had OCC data). Besides the measurement of SCC, there are several additional sensor technologies for each manufacturer on the market that could have potentially improved the accuracy of mastitis detection, such as lactate dehydrogenase detection [60,61] or rumination activity [62,63]. However, these technologies were not sold as a standard part of the AMS mastitis detection sensor package [64]. Therefore, they were not evaluated in this study. Furthermore, we assessed the standard udder health alert lists of each system based on discussions with manufacturer personnel. Unfortunately, one Lemmer-Fullwood list (“control_report_milking_10 days”) could not be used due to missing headers. However, the data might have been indicative of udder health problems, as they included cow activity [65] and milk lactose [66]. Due to these limitations, one has to assume that when considering the stated and theoretically available additional sensor data the SN and SP could have actually been higher than we were able to determine with the given data. Likewise, it must be considered that farmers using a Lely or Lemmer-Fullwood AMS could adjust the limits of the pre-installed warning lists. A check of the settings at the time of the data backup was carried out, but due to this technically conditioned snapshot at the herd visit, an adjustment of the threshold values shortly before cannot be ruled out. For this reason, possible herd-specific limit value adjustments were not taken into account and a possible improvement or deterioration of the SN and SP with farm-specific values could not be evaluated.

The low specificity estimates of this study highlight a known problem [67]. A high number of false positive alerts as a result of low prevalence of CM and a lower SP of the detection system can lead to economic losses due to wrongly discarded milk [67] and increased labor cost to assess each of these animals’ udder health in the barn. This will decrease the farmer’s confidence in the AMS’ udder health alerts [68]. The low SP may be due to several factors: first, CM is a dynamic event that passes through different stages [69], second, the AMS alerts do not distinguish the grade of mastitis (e.g., subclinical or clinical), and third, the manufacturers did not provide system definitions for clinical mastitis. Therefore, this purely binary classification may lead to higher false positive rates since marked cows may not yet have an apparent change in milk but may already show changes in milk components detectable by the sensors [33,46]. The gold standard definition of ISO 20966:2007 also proposed by Kamphuis et al. [44,70], i.e., the presence of clots or flakes in two out of three consecutive milkings, thus tries to counteract this and capture the evolution process of CM. The single assessment of the udder in our study was selected for several reasons: (1) the farmer has to assess cows on the lists at least twice a day and new mastitis cases should be present; (2) the visual assessment allowed for the assessment of the quality of visually abnormal milk, watery character of the milk, or single abnormal milk [71]. Therefore, we are closer to the basis of legal assurance of milk fit for human consumption (EU Directive EC/853/2004) based on the organoleptic examination of milk and udder pre-milking by milkers [43,72]. The main concern was that none of the manufacturers provided a definition of “clinical mastitis” that would allow for a simple “yes/no” answer. Solutions to this dilemma could be probability indications of potential clinical mastitis as suggested by Friggens et al. [69] and different alerting approaches for different forms of mastitis or udder health situations [45]. We were able to identify several automatically collected parameters that were associated with SN and SP in multivariable regression models. These factors may be used by the farmer to improve CM detection. However, although EC was included in many algorithms already, the additional consideration of high EC improved the identification of CM by DeLaval and GEA systems. This suggests that cows with CM have higher milk EC values [73] and the effects of, for example, temperature, fat, and milk fractions on EC [18,52,53] play a lesser role in EC than the underlying inflammation [20]. Due to the relatively small number of CM cases, only a few additional predictors could be identified in the SN models, and none for Lemmer-Fullwood. In contrast, several parameters associated with SP were identified, which can partly be attributed to the substantially larger sample size. For example, it has been found that cows with longer milking intervals in DeLaval and GEA systems have a lower chance of being correctly classified as healthy cows. This could be because the likelihood of a cow having abnormal milk increases as the milking interval increases [74], and the flakes that may be present may affect the measurement of EC [18], making it difficult to alert correctly; additionally, a milking visit could be protracted due to pain caused by the onset of udder inflammation.

The one factor that was consistently helpful across all systems was the trend of SCC between the last two monthly milk tests. Cows with “chronically” high SCC (i.e., two subsequent tests with SCC > 100.000 cells/mL) had lower chances of being correctly classified as healthy by the AMS. One explanation could be that “chronic” cows had persistent or repeated subclinical mastitis and this triggered the alert by the AMS. As there is no specific alert for CM — only udder health “abnormalities”, this may drive the misclassification for clinical mastitis [45]. Regular checking of these cows with repeated high SCC values should be performed to detect CM early on the one hand, and to prevent unnecessary automatic separation of milk by the AMS on the other hand. Although the models included a large number of automatically recorded parameters, they did not include all possible parameters, especially in the Lemmer-Fullwood data set. Similarly, farm-specific factors such as setting alert limits were not part of the available data. The herd-level random effect accounts for some differences among these unknown factors. Inclusion of additional interactions in the model could have potentially improved the model fit. However, due to the lack of biologically plausible interactions to be considered and the low sample size in all SN models they were not further explored.

Although AMSs sufficiently identify cows with clinical mastitis, farmers must continue to know and monitor their animals several times per day in the barn. Otherwise, cows that were unable to visit the AMS due to acute severe disease (e.g., mastitis), might be identified too late if flagged simply based on milking interval, or the milk of cows with CM not yet identified by the machine could enter the food supply as well. On the other hand, the milk of healthy cows would be discarded longer than necessary, if the farmer does not assess the udder status of the cow. While the benefits of an AMS are more flexible working hours and a daily overview over each animal’s health and production data [38,64], the farmer needs to understand the provided data and machine functions and evaluate their animals in the barn to ensure food safety and animal health. In addition, AMS producers need to revise their approaches to detect udder health problems, especially CM. For example, there are promising studies on intelligent data processing and machine learning to use the amount of data collected to identify sick cows [75,76,77]. In addition, new sensor technologies closer to the origin of the gold standard definition, the organoleptic (especially visual) inspection of the milk, as well as information from other sensors that measure other more general health parameters should be explored. The former could include camera-based milk quality assessment of the fore stripping. These technologies are already being used in other livestock production chains, for example, to detect and assess footpad health in poultry [78]. Until then, a more comprehensive implementation of DHIA data in herd management software along with AMS data may be helpful for monitoring udder health.

## 5. Conclusions

The present study shows the SN and SP of CM detection from different AMSs used in Bavaria. Overall, the detection of clinical mastitis by different AMSs was found to be sufficient, but the low specificity could cause unnecessarily discarded milk and additional workload for farmers to check on their animals. Some automatically collected parameters, such as EC and monthly test day results, are related to the current detection performance of CM by AMSs and can be helpful to farmers in their assessment of AMS udder-health alerts. Because there is currently no official definition of visibly abnormal milk (i.e., clinical mastitis) by the manufacturers, farmers need to consider test day data and results of udder health evaluations of each cow in the barn to interpret and act on AMS lists appropriately to warrant food safety, milk quality, animal welfare, and the economic viability of their farm.

## Figures and Tables

**Table 1 animals-12-02131-t001:** Udder health lists of each manufacturer’s herd-management software used for this study.

AMS/Software	Lists	Content and Explanation
DeLaval/ DelPro Farm Manager 5.5	cow_monitoring ^1^	Sensor (e.g., EC ^2,^*, MY ^3,^*, blood occurrence *, etc.) and cow data (e.g., MDi ^4,^**, MI ^5,^, DIM ^6^ etc.).
	Milking_data_last_30_days ^1^	Sensor and cow data of the last 30 days.
GEA Farm Technologies/Dairy Plan C21	Daily_checked_lists ^1^	Summary of lists to be checked daily in the program. Indicates whether cows appear on these lists or not (1/0).
	AMS_udder_health_monitoring-List1	Displays cows with a deviation in EC ^2^ value between the quarter with the highest average EC^2^ value and the quarter with the lowest average EC^2^ value in the last 4 milkings.
	AMS_udder_health_monitoring-List2	Displays cows with an EC ^2^ deviation within a quarter.
	AMS_udder_health_monitoring-Acute_warnings	Summarized cows which have been flagged on both List1 and List2.
	AMS_increased_conductivity	Displays cows with an EC deviation within a quarter.
	Mrobot_milk_decline	Displays cows with a milk decline.
	Mrobot_to_be_milked	Displays cows overdue for milking.
	Herd_status_current_last_milking ^1^	Sensor (e.g., EC ^2,^*, MY ^3,^*, blood occurrence *, MT ^7,^* etc.), and cow data (e.g., MI ^5^, DIM ^6^ etc.).
	Milking_data_for_the_last_10_days ^1^	Sensor and cow data of the last 10 days.
	QuarterCellCount_alert	Alert list using SCC ^8^*.
Lely/ T4C-Time for cows	Dailymilkproduction ^10^	Sensor (e.g., MYD ^9,^**, milk fat **, milk protein **, etc.) and cow data (e.g., feed intake, DIM ^6^, etc.) of current milking.
	Milkings_last_7 days ^10^	Sensor (e.g., EC ^2,^*, milk color *) and cow data (e.g., DIM ^6^ etc.) of the last seven days.
	Action_list ^10,11^	Displays cows and their sensor data with a new indication such as MYD ^9,^**, EC ^2,^*, MT ^7,^**, SCC ^8,^** for 24 h on this list.
	Monitor_list ^10,11^	Displays cows until a milking without mastitis indicator (MYD ^9^, EC ^2,^*, MT ^7,^**, SCC ^8,^**) occurred.
Lemmer-Fullwood/ Chrystal Fusion	Control_report_10_days ^10^	Sensor (e.g., milk protein, milk fat, lactose, etc.) and cow (e.g., DIM, MI) data of the last 10 days including an alert for suspected mastitis, based on EC ^2,^*.
	Kick_off ^10^	Kick-off event (yes/no) of the teat cups per quarter of the last 10 days.
	4qcm_10_days ^10^	Displays data of EC ^2,^* at quarter level for each milking of the last 10 days.
	Control_report_milking ^10^	Displays cows for udder health monitoring.

^1^ Study lists created in cooperation with employees of the respective manufacturers on the basis of herd management program lists; ^2^ EC = Electrical conductivity of milk (manufacturer’s internal unit); ^3^ Milk yield (kg) ^4^ MDi = Mastitis detections index. The MDi is a mathematically generated index that considers EC and blood presence (both measured at quarter level) as well as milking interval; ^5^ MI = Milk interval; ^6^ DIM = days in milk; ^7^ MT = milk temperature, (°C); ^8^ SCC = somatic cell count, (manufacturer’s internal unit); ^9^ MYD = Milk yield per day, (kg); ^10^ Created by the milking equipment service team for this study based on originally used lists; ^11^ Original lists, modified by software service staff of the companies to show all lactating cows, but marked cows which were originally indicated on each list; * = at quarter level; ** = at cow level.

**Table 2 animals-12-02131-t002:** Overview of raw and generated cow test day data from the regional Dairy Herd Improvement Association (DHIA) for this study on AMS system accuracy in Bavarian dairy herds.

Source	Variable
Test day data	Cow identification
	Date of birth
	Breed
	Lactation number
	Days in milk at the test day
	Date of monthly test day
	Milk yield (kg)
	Fat (%)
	Protein (%)
	Urea concentration (ppm)
	SCC (cells/mL)
Generated ^1^	Test day with SCC ≥ 700,000 cells/mL (1/0)
	Number of test days with SCC ≥ 700,000 cells/mL (*n*)
	Test day with SCC ≥ 400, 000 cells/mL (1/0)
	Number of test days with SCC ≥ 400, 000 cells/mL (*n*)
	Number of missing test day data (*n*)
	Udder health status (categorization, based on two subsequent test days in that lactation):
	chronic: two subsequent tests with >100.000 cells/mL
	new IMI ^2^: previous SCC < 100.000 and current SCC > 100.000 cells/mL
	cured: previous SCC > 100.000 and current SCC < 100.000 cells/mL
	healthy: both tests < 100.000 cells/mL
	no current test data: only data of 1 test day available
	no DHIA data available

^1^ New variables were generated after the data for monthly test days from cows with DIM ≤ 5 (no measurement) and a dry period or both were excluded; ^2^ IMI: Intramammary infection.

**Table 3 animals-12-02131-t003:** Overview of data-cleaning process in a study on AMS system accuracy in Bavarian dairy herds.

		DeLaval	GEA	Lely	Lemmer-Fullwood	Overall
Study herds, *n*		27	29	31	27	114
AMS data	Backup at farm visit, *n* (restored ^1^)	20 (7)	29	31	26 ^2^	113
	Last milking data, *n*	2047	1721	2247	1974	7989
Evaluated cows at farm visit, *n*		1904	1664	2152	1691	7411
Cows excluded due to, *n*	Incorrect identification	13	8	21	31	73
Not matching with AMS data		12	6	16	68	102
DIM < 3d		22	21	18	11	72
>24 h since last milking		11	9	11	6	37
3-teater cows, i.e., quarter with CM ^3^ not milked by AMS		15	4	10	8	37
No alert information available		-	-	-	22	22
Cows in final statistical analysis, *n*		1831	1616	2076	1545	7090
Additional DHIA ^4^ Data		1665	1462	1879	1534	6540
Last three test day data available		1517	1309	1636	1425	5887
Only last test day data available		99	107	156	45	407
No test day data available		166	154	197	33	550
Cows with CM ^3^, *n* (affected quarters, *n*)		70	54	78	37	239
Grade 1—mild: abnormal milk		60 (62)	52 (59)	69 (76)	31 (42)	212 (239)
Grade 2—medium: abnormal milk and/or swollen quarter		9 (10)	2 (2)	8 (8)	5 (5)	24 (25)
Grade 3—Severe: grade 1 or 2 with systemic signs		1	-	1	1	3

^1^ AMS Data restored of the automated daily backups from 1 or 2 am; ^2^ AMS data of one Lemmer-Fullwood herd could not be restored; ^3^ CM = clinical mastitis; ^4^ DHIA = regional Dairy Herd Improvement Association.

**Table 4 animals-12-02131-t004:** Overview of the characteristics of participating herds summarized per manufacturer. Farm visits between September 2019 and August 2020 for a study on AMS system accuracy in Bavarian dairy herds. Unless otherwise stated, the median (25th–75th percentile) is reported.

Characteristic		DeLaval	GEA	Lely	Lemmer-Fullwood	Overall
Participating herds, *n*		27	29	31	27	114
Number of AMSs, *n*		31	30	35	30	126
Year of AMS installation,	median	2014	2018	2015	2017	2017
(min–max)		(2007–2020)	(2016–2020)	(2009–2019)	(2011–2019)	(2007–2020)
Herd size ^1^	mean, ±SEM	71 ± 4.8	57 ± 3.1	69 ± 4,3	63 ± 2.9	65 ± 1.9
(min–max)		(40–139)	(28–106)	(35–127)	(31–100)	(28–139)
Herd average milk yield ^2^, kg		8424 (7905–8875)	7700 (7126–8573)	8949 (8515–9325)	8400 (7981–9106)	8525 (7700–9135)
Bulk tank ^3^ (×10^3^/mL)						
Somatic cells/mL		176	126	202	130	155
		(140–240)	(103–154)	(155–241)	(101–178)	(124–210)
Bacterial count, cfu/mL		13	17	12	17	15
		(10–19)	(11–25)	(9–17)	(13–26)	(10–21)
Clinical mastitis prevalence ^4^, %		4.1	2.7	3.8	2.9	3.4
Herds without clinical mastitis, *n*		3	4	3	6	16
Operating structure, % herds						
Conventional		85	86	97	93	90
Organic		15	14	3	7	10
DHIA ^5^ member		96	97	97	100	
Breed, % herds						
Simmental		19	86	58	82	61
Mixed		37	7	23	11	19
Brown Swiss		26	3	10	7	11
Other (incl. Holstein Friesian)		19	4	10	-	8
Period of the farm visits (2019–2020)		Oct–Aug	Apr–Aug	Sep–Mar	Feb–Aug	Sep–Aug

^1^ Number of lactating cows, ^2^ Result of herd performance (365 d)/number of tested cows, ^3^ Data of last available bulk tank analysis, ^4^ Median intra-herd prevalence of herds with cows with clinical mastitis, ^5^ DHIA = Dairy herd improvement association is the Landeskuratorium der Erzeugerringe für tierische Veredelung in Bayern e.V.

**Table 5 animals-12-02131-t005:** Sensitivity and specificity of clinical mastitis (CM) alerts for AMSs from DeLaval, GEA, Lely and Lemmer-Fullwood.

AMS	Cows, *n*	CM, *n* Cases	AMS Alert Used		Sensitivity, %	95% CI, % ^1^	Specificity, %	95% CI, % ^1^
DeLaval	1831	70	MDi ^2^	≥1.4	61.4	49.0–72.8	89.3	87.7–90.7
			≥2.0		31.4	20.9–43.6	97.2	96.3–97.9
GEA	1616	54	AMS_udder_health_monitoring ^3^					
			List1 ^4^		75.9	62.4–86.5	79.2	77.1–81.2
			List2 ^5^		48.2	34.3–62.2	93.5	92.1–94.6
			Acute_Udder_Health_warnings ^6^		38.9	25.9–53.1	94.9	93.7–95.9
Lely	2076	78	Monitor_list ^7,9^		78.2	67.4–86.7	86.2	84.6–87.7
			Action_list ^7,9^		28.2	18.6–39.5	94.9	93.8–95.8
Lemmer-Fullwood	1545	37	4QCM alert ^8,9^		67.6	50.2–82.0	92.2	90.8–93.5

^1^ Exact confidence interval, ^2^ MDi = Mastitis detections index. The MDi is a mathematically generated index that considers EC and blood presence, which are both measured at quarter level and milking interval. It uses values between 1 and 6; ^3^ CM alert list based on deviations in EC. This list is divided into three subgroups: List1, List2 and Acute_Udder_Health_warnings; ^4^ List1: displays cows that have a deviation of ≥30% in EC value between the quarter with the highest average EC value and the quarter with the lowest average EC value in the last 4 milkings. The lowest value must be greater than 400 (manufacturer’s own unitless measurement), ^5^ List2: displays cows with an EC deviation within a quarter of the default setting of ≥110%; ^6^ Acute_Udder_Health_warnings: displays cows which are indicated on List1 and List2; ^7^ Two udder health lists, modified from the pre-installed lists Report12 and Report23 in the system to show all cows, but those cows originally shown on these lists were marked. Action_list, e.g., Report12, displays cows with a new indication for 24 h on this list, while on Monitor_list, e.g., Report23, cows remain on this list until a milking without indication occurs. Factory settings of the indication limits for these reports are: milk yield deviation of daily milk production of 4.0 kg or 20%; decrease of daily milk production of more than 7 kg; EC deviation from last milking of 20%, EC deviation from the 3-day average of 20%; absolute EC threshold of 100; milk temperature above a deviation factor of 2.0%; exceeding the SCC threshold of 500, measured by the MQC-C system; ^8^ 4QCM system (quarter conductivity measurement system): measures the EC at the quarter level and with standard limits for the conductivity per quarter; deviation by 35% from the 10-day average and threshold value of the measured value of >7.5 mmho. ^9^ CM alert thresholds could be customized by farmers.

**Table 6 animals-12-02131-t006:** Factors improving the sensitivity of clinical mastitis detection (outcome: true positive) in a cow-level multivariable logistic regression analysis including automatically recorded data from AMS and DHIA test days. Only cows with clinical mastitis were used and herd was included as random effect.

AMS	CM, *n*	AMS Alert	Predictor	β ^1^	SEM	Odds Ratio	95% CI	*p*-Value
DeLaval	70	MDi ^2^ ≥ 1.4	Intercept	−3.56	1.76			0.05
			EC of current milking	0.22	0.10	1.24	1.03–1.51	0.03
GEA	54	List 1 ^3^	Intercept	−4.43	1.77			0.02
			ΔQEC ^4^	0.03	0.01	1.03	1.01–1.05	0.01
Lely	78	Monitor list ^5^	Intercept	−7.56	2.62			0.01
			LogSCC ^6^	1.41	0.42	4.10	1.75–9.52	<0.01
Lemmer-Fullwood	37	4QCM ^7^ Alert	-	-	-	-	-	-

^1^ β = Regression coefficient; ^2^ MDi = Mastitis detections index. The MDi is a mathematically generated index that considers EC and blood presence, which are both measured at quarter level and milking interval. It uses values between 1 and 6; ^3^ List1: a warning list for udder health problems and displays cows that have a deviation of ≥30% in EC value between the quarter with the highest average EC value and the quarter with the lowest average EC value in the last four milkings. The lowest value must be greater than 400 (manufacturer’s own unitless measurement); ^4^ ∆QEC = Difference of the highest to lowest quarter EC measurement; ^5^ Monitor list, e.g., Report23, is a warning list for udder health problems and displays cows until a milking without indication occurs. Factory settings of the indication limits for these reports are: milk yield deviation of daily milk production of 4.0 kg or 20%; decrease of daily milk production of more than 7 kg; EC deviation from last milking of 20%, EC deviation from the 3-day average of 20%; absolute EC threshold of 100; milk temperature above a deviation factor of 2.0%; exceeding the SCC threshold of 500, measured by the MQC-C system; ^6^ SCC determined by the Lely MQC-C System, log transformed; ^7^ 4QCM system (quarter conductivity measurement system): measures the EC at the quarter level and with standard limits for the conductivity per quarter; deviation by 35% from the 10-day average and threshold value of the measured value of >7.5 mmho (manufacturer-specific unit).

**Table 7 animals-12-02131-t007:** Factors improving the specificity (outcome: healthy cows, i.e., no clinical mastitis, without alert) on a cow level, including only healthy cows and DHIA as well as AMS data, in a cow-level multivariable logistic regression analysis with herd as random effect.

AMS, *n* Cows	Predictor		β ^1^	SEM	Odds Ratio	95% CI	*p*-Value
DeLaval	Intercept		−1.93	2.11			0.37
1761	Milking interval in hours		−0.20	0.03	0.82	0.78–0.88	<0.01
	Δ highest and lowest quarter EC ^2^		−0.59	0.10	0.56	0.46–0.67	<0.01
	DHIA ^3^ lactose concentration		1.60	0.42	4.94	2.16–11.29	<0.01
	Udder health status ^4^	chronic	−1.70	0.27	0.18	0.10–0.30	<0.01
	new IMI ^5^		−1.11	0.34	0.33	0.17–0.64	<0.01
	cured		−0.20	0.44	0.82	0.35–1.93	0.65
	no DHIA data		−0.83	0.61	0.44	0.13–1.46	0.18
	no current test data		−0.64	0.67	0.53	0.14–1.96	0.34
	healthy		Referent				
GEA	Intercept		1.72	0.28			<0.01
1562	Milk yield of last milking, kg		0.01	0.02	1.12	1.08–1.17	<0.01
	Milking interval in hours		−0.10	0.02	0.90	0.87–0.93	<0.01
	Lactation number	1	0.60	0.17	1.82	1.30–2.55	<0.01
	2		0.11	0.16	1.11	0.81–1.54	0.51
	≥3		Referent				
	Udder health status ^4^	chronic	−1.64	0.20	0.19	0.13–0.29	<0.01
	new IMI ^5^		−1.46	0.23	0.23	0.15–0.36	<0.01
	cured		−0.74	0.25	0.48	0.29–0.79	<0.01
	no DHIA data		−1.03	0.24	0.37	0.23–0.58	<0.01
	no current test data		−0.98	0.34	0.38	0.19–0.73	<0.01
	healthy		Referent				
Lely,	Intercept		8.89	0.72			<0.01
1998	Quarter based EC threshold of 72	0	1.32	0.20	3.73	2.54–5.47	<0.01
	1		Referent				
	Fat content (measured by AMS)		−0.31	0.10	0.74	0.60–0.90	0.03
	LogSCC ^6^		−1.19	0.12	0.30	0.24–0.38	<0.01
	Udder health status ^4^	chronic	−0.80	0.25	0.45	0.27–0.73	<0.01
	new IMI ^5^		−0.33	0.33	0.72	0.38–1.38	0.32
	cured		−0.57	0.38	0.95	0.45–2.01	0.88
	no DHIA data		−0.68	0.38	0.51	0.24–1.06	0.07
	no current test data		−0.09	0.45	0.91	0.38–2.21	0.84
	healthy		Referent				
Lemmer-Fullwood, 1508	Intercept		1.19	0.33			<0.01
	Quarter based EC threshold of 5.6	0	2.58	0.32	13.13	7.03–24.51	<0.01
	1		Referent				
	Udder health status ^4^	chronic	−1.60	0.31	0.20	0.11–0.37	<0.01
	new IMI ^5^		−0.35	0.45	0.71	0.29–1.72	0.44
	cured		−0.40	0.55	0.67	0.23–1.97	0.47
	no DHIA data		−0.20	0.60	0.82	0.26–2.63	0.74
	no current test data		−0.80	0.58	0.45	0.14–1.41	0.17
	healthy		Referent				

^1^ β = Regression coefficient; ^2^ EC: Electrical conductivity measurement; ^3^ DHIA: Dairy herd improvement association is the Landeskuratorium der Erzeugerringe für tierische Veredelung in Bayern e.V.; ^4^ Udder health status: Categorization, based on DHIA somatic cell count measurement (SCC). SCC from two subsequent test days in that lactation with a cutoff value of 100,000 cl/mL are compared; ^5^ IMI: Intramammary infection; ^6^ SCC determined by the Lely MQC-C System, log transformed.

## Data Availability

The data of this study are property of the Bavarian Animal Health Services. None of the data were deposited in an official repository.

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
