# Peer review of "Sensitivity and Specificity for the Detection of Clinical Mastitis by Automatic Milking Systems in Bavarian Dairy Herds"

_animals, 2022, doi:10.3390/ani12162131_

Round 1
Reviewer 1 Report
I found the paper to be very well written and almost free of spelling, and grammar errors etc. The only two errors I noted were und instead of and on line 454 and septation instead of separation on line 555. I have only one major concern with the paper and that is the interpretation of SN and SP. On lines 38 through 42 the numbers show SN percentages for all four brands that are below 80% which I believe Hoogeveen et. al. suggested as a reasonable standard. But the paper says the "minimum SN limit" of 80% was met by three of the brands and reports data where the top of the confidence interval is higher than 80%. I think the confidence interval means that the final number that would result from more data has a 95% likelihood of falling between the two limits, but that is no justification for claiming the higher limit as the actual SN. I am not a statistician so the fact I am questioning this may just reflect my own ignorance. I am seeking help with my understanding of this method of reporting SN data but have no found an answer in the time allowed for review so all I can ask is that this be better explained to us readers with limited statistical knowledge. One final minor point is that you make a lot of use of acronyms and I think that is useful when a term is repeated 10 or more times, but you acronym Bavarian Animal Health Services with TGB and then just refer to them one more time in the paper. I don't think that is the intent of an acronym. Even something like GS for gold standard is probably overkill. It isn't that hard to just write gold standard the four times it appears in the text.
Reviewer 2 Report
Dear Authors,
I have studied the work thoroughly and I come to the following remarks:
1- My recommendation for changing the subject to “Sensitivity and specificity for the detection of subclinical mastitis by automatic milking systems in Bavarian dairy herds. The parameters studied are well suited for detection at the stage of subclinical mastitis, in order to quickly identify the situation and solve the problems. There are several parameters for clinical mastitis and what is shown in the animal, such as: increased temperature, udder reddening, udder swelling, reduced milk yield, reduced feed intake and the animals may not be able to milk. These parameters are clearly visible and indicate to farmers that the animal has clinical mastitis.
2- The authors have already shown on my recommendation in the abstract: cows that were above the threshold of 100,000 cells/ml for subclinical mastitis in both test day had lower chances of being classified as healthy by the AMS compared to cows that were below the threshold. We know that the SCC in milk is up to 100,000 cells/ml as a physiological level and above this value it is called subclinical mastitis because milk up to 300,000 or 400,000 cells/ml is sold to the dairy without any problems.
3- In line 39, the abbreviation CI must be clarified.
4- In line 155 the reference: BRADLY et GREEN [40] must be spelled correctly, as BRADLY and GREEN [40].
5- In line 171 and table 1, the authors did not mention which sensors are used in the AMS-DeLaval.
6- In line 158, AMS data, the authors did not mention that each sensor is for each parameter and the question is, e.g., at Lely company: are there three sensors for three measured parameters (SCC, MT and EC) in each quarter?
7- In line 452, the authors mentioned that if the farmer relied solely on the system warnings, this would result in improper disposal of significant quantities of milk and associated economic losses. This contradicts what was previously written that the AMS can show good information for clinical mastitis. Is there an explanation for this?
8- In line 536, is the milk percentage the same milk fractions as mentioned in line 469? or not.
9- In lines 563 and 564, the authors mentioned that due to the improvable performance, especially SP, of CM detection by AMS, humans remain an important factor in ensuring udder health and food safety on dairy farms. Since the human being is the most important factor in the detection of clinical mastitis, such expensive measures in the AMS, which end up providing incorrect information, are not necessary.
10- In line 581, the authors mentioned that the EC and monthly test daily results are related to the current detection performance of CM by AMS and can help farmers to assess AMS udder health. But in line 468 the authors mention that EC alone is not considered sufficient for CM detection. Is there an explanation for this?
11- There are a few notes in the reference list:
1) In the following references (20, 32, 41, 50, 65, 77), are without complete page numbers at the end of the studies.
2) Reference 22, the country for the conference must be mentioned.
3) The reference 42 should be mentioned whether it is a book or a journal.
I wish you much success
Reviewer 3 Report
The paper is interesting with practical output. The detection of clinical mastitis by automatic milking system is an important problem in dairy farms. The authors describe well this topic and evaluate different systems for the detection of clinical mastitis.
The material and methods describe the gold standard definition – “Based on the findings from clinical examination of the udder, CM was recorded as grade 1 (abnormal milk with change in character, like watery or bloody, and or the occurrence of flakes, clots or pus of at least one quarter), grade 2 (abnormal milk in addition to local signs of inflammation of the quarter), or grade 3 (signs of grades 1 and 2 in addition to systemic signs, e.g., off feed, fever) in agreement with BRADLY et GREEN [40]. The gold-standard (GS) definition of CM in our study included the grades 1-3.“
Did the authors use in the evaluation also grades of clinical mastitis?
Did the authors consider the number of affected quarters for the evaluation?
Abstract / Conclusion
Line 38 – 42 „All but one manufacturer achieved the minimum SN limit of > 80% (DeLaval (SN: 61.4% (95% CI: 49.0%- 72.8%)), GEA (75.9% (62.4%-86.5%)), Lely (78.2% (67.4%-86.8%)), and Lemmer-Fullwood (67.6% (50.2%- 82.0%)). However, none of the evaluated AMSs achieved the minimum SP limit of 99% (DeLaval (SP: 89.3% (95% CI: 87.7%-90.7%)), GEA (79.2% (77.1%-81.2%)), Lely (86.2% (84.6%-87.7%)), and Lemmer-Fullwood (92.2% (90.8%-93.5%)).“
Line 578-579 „Overall, the detection of clinical mastitis by different AMS was found to be sufficient, but the low specificity could cause unnecessarily discarded milk and additional workload to farmers to check on their animals.“
I don't think this conclusion is entirely correct. It is true that three of the four manufactures were able to achieve the minimum SN limit of 80%, when considering 95% CI. However, the confidence intervals are really large and in some cases the lower value is very low. Is it really true that sensitivity of all systems is sufficient, as you write in the conclusion?
I agree that low specifity could cause unnecessary wasted milk and additional workload for farmers, and thus economic losses. Nevertheless, low sensitivity is dangerous in terms of milk quality and human health.
